# Neuromodulation of excitatory synaptogenesis in striatal development

Yevgenia Kozorovitskiy[1,2], Rui Peixoto[1], Wengang Wang[1], Arpiar Saunders[1†], Bernardo L Sabatini[1]*

[1]Department of Neurobiology, Howard Hughes Medical Institute, Harvard Medical School, Boston, United States; [2]Department of Neurobiology, Northwestern University, Evanston, United States

**Abstract** Dopamine is released in the striatum during development and impacts the activity of Protein Kinase A (PKA) in striatal spiny projection neurons (SPNs). We examined whether dopaminergic neuromodulation regulates activity-dependent glutamatergic synapse formation in the developing striatum. Systemic in vivo treatment with $G\alpha_s$-coupled G-protein receptors (GPCRs) agonists enhanced excitatory synapses on direct pathway striatal spiny projection neurons (dSPNs), whereas rapid production of excitatory synapses on indirect pathway neurons (iSPNs) required the activation of $G\alpha_s$ GPCRs in SPNs of both pathways. Nevertheless, in vitro $G\alpha_s$ activation was sufficient to enhance spinogenesis induced by glutamate photolysis in both dSPNs and iSPNs, suggesting that iSPNs in intact neural circuits have additional requirements for rapid synaptic development. We evaluated the in vivo effects of enhanced glutamate release from corticostriatal axons and postsynaptic PKA and discovered a mechanism of developmental plasticity wherein rapid synaptogenesis is promoted by the coordinated actions of glutamate and postsynaptic $G\alpha_s$-coupled receptors.

*For correspondence: bernardo_sabatini@hms.harvard.edu

Present address: †Department of Genetics, Harvard Medical School, Boston, United States

Competing interests: The authors declare that no competing interests exist.

## Introduction

In the vertebrate basal ganglia, dopamine performs critical functions in motivated, goal-directed learning and behavior, transmitting signals related to rewarding stimuli and other salient experiences (*Bromberg-Martin et al., 2010*). Classical models of dopaminergic signaling in adult animals propose that the activity of dopamine-producing neurons encodes reward prediction errors and that released dopamine regulates the strength of corticostriatal excitatory glutamatergic inputs. This process is thought to favor the execution of a specific action that previously led to reward, at the expense of competing motor programs. An important element of this model is the distinct action of dopamine on the direct and indirect pathways, comprised of two classes of striatal spiny projection output neurons that express different types of dopamine receptors, project to separate downstream targets, and regulate complementary aspects of motor behavior (*Bateup et al., 2010*; *Kravitz et al., 2010*; *Cui et al., 2013*). Dopamine differentially regulates biochemical signaling (*Nishi et al., 1999*; *1997*), neuronal activity and synaptic plasticity in these two classes of striatal neurons, promoting synaptic potentiation and depression through different receptor subtypes (*Shen et al., 2008*).

Less is known about the function of dopamine in the postnatal development of basal ganglia, when goal-oriented motor programs are first expressed and activity-dependent synapse formation occurs in the striatum. Therefore, we investigated whether dopamine modulates activity-dependent striatal plasticity and synapse formation during this time. In mice, the second postnatal week of life is a period of rapid growth of excitatory synapses on SPNs, a process driven by the release of glutamate from corticostriatal axons (*Kozorovitskiy et al., 2012*), among other striatal glutamatergic projections. Furthermore, the balance of activity in direct and indirect pathway SPNs (dSPNs and iSPNs,

**eLife digest** The brain is composed of intricate circuits of connected neurons that communicate via a combination of electrical and chemical signals. Some signals (referred to as excitatory signals) increase the probability that the neuron receiving the chemical message will produce an electrical impulse. On the other hand, inhibitory messages decrease the likelihood of this activity. Both of these kinds of signals are fast, and act over milliseconds. There is also a diverse set of slower signals, referred to as neuromodulation, which regulates the faster signals. A signaling chemical called dopamine is involved in neuromodulation and is essential for rewarding behavior and complex motor actions. The importance of dopamine is clear from the profound lack of movement seen in individuals with Parkinson's disease, which is caused by the death of dopamine producing brain cells.

Many nerve endings from dopamine-releasing neurons connect to a part of the brain's reward system called the striatum. The neurons in this region are organized into two pathways that have opposing impacts on behavior. Dopamine activates different kinds of receptors called "G protein-coupled dopamine receptors" on neurons from each pathway. This allows dopamine to alter the activity of a protein called Protein Kinase A (or PKA) and alter the signaling state of these neurons.

The impact of dopamine on neural circuits in adults has been extensively studied. However it was unknown whether dopamine might influence how neural circuits are wired during brain development. Because the nerve endings from dopamine-releasing neurons reach the striatum before most excitatory connections between the neurons are formed, dopamine stands to influence the development of connections in the striatum.

Kozorovitskiy et al. have now investigated the role of neuromodulation in brain development in young mice. This involved measuring the formation of excitatory connections or synapses and the electrical activity of different striatal neurons during the maturation of brain circuits that occurs after birth. This analysis revealed that turning on dopamine receptors that increase PKA activity rapidly enhances the number of excitatory synapses on the neurons that express this receptor.

Kozorovitskiy et al. then used a variety of approaches to investigate whether there is cooperation between G protein-coupled receptors, PKA activity and a signaling molecule called glutamate in striatal development. This revealed a more general mechanism by which the activation of G-protein-coupled receptors interacts with glutamate (the primary excitatory signal sent between neurons) in order to produce new synapses. These results reveal a previously unknown role for neuromodulation in "wiring up" the brain and open the possibility of new therapies to treat neurodevelopmental and neurodegenerative disorders.

respectively) influences activity in the neocortex, potentially altering the firing patterns of cortico-striatal neurons via recurrent circuit interactions (*Kozorovitskiy et al., 2012*; *Oldenburg and Sabatini, 2015*). Thus, positive feedback in recurrent circuits can amplify early glutamatergic signaling, rapidly changing network connectivity. If dopamine modulates glutamatergic synapse function in developing SPNs as it does in mature circuits, it likely interacts with glutamate-dependent neural circuit development during the early postnatal period. In dSPNs, dopamine acts on type 1 dopamine receptor (Drd1), a $G\alpha_s$-coupled GPCR to activate PKA, a critical enzyme involved in cell growth and plasticity (*Nishi et al., 2000*). In contrast, in iSPNs, dopamine activates type 2 dopamine receptors (Drd2), $G\alpha_i$-coupled GPCRs, decreasing PKA activity. Since PKA activity levels influence spinogenesis and synaptogenesis in cortical pyramidal neurons (*Kwon and Sabatini, 2011*), dopamine may directly set the threshold for activity-dependent synaptogenesis in striatal SPNs.

Dopamine is poised to regulate striatal signaling early in postnatal life, since embryonic striatum receives substantial dopaminergic innervation from substantia nigra pars compacta (SNc) and the ventral tegmental area (VTA), as is evidenced by neuroanatomical studies (*Hu et al., 2004*) and direct functional assays showing embryonic dopamine release (*Ferrari et al., 2012*). Two observations indirectly support the involvement of dopamine in the developmental wiring of the striatum. First, dysfunctions of the dopaminergic system are linked to numerous diseases that are postulated to have a developmental origin, including schizophrenia (*Lau et al., 2013*; *Laruelle, 2014*), obsessive compulsive disorder (*Nikolaus et al., 2010*) and Tourette Syndrome (*Buse et al., 2013*). Second,

embryonic exposure to drugs of abuse that act on the dopaminergic system, such as methamphetamine (*Heller et al., 2001*), induces profound brain and behavioral abnormalities in the offspring.

Here we tested the hypothesis that neuromodulators, and specifically dopamine, regulate glutamate-dependent synaptogenesis during striatal development. We found a potentially general mechanism of developmental synaptogenesis that relies on the coordinated actions of glutamate and the activation of PKA via G$\alpha_s$-coupled receptors.

## Results

Consistent with early development of the dopaminergic projection from SNc/VTA to striatum, the dopamine transporter promoter is active and dopaminergic axons are found in the neonatal striatum on postnatal day 1 (P1) (*Figure 1A*). By P10 the axons of midbrain dopaminergic neurons densely innervate the striatum and express tyrosine hydroxylase, a critical enzyme for dopamine synthesis (*Figure 1B*), indicating that dopaminergic neuromodulation is poised to regulate synapse and cell function during this early developmental stage.

To address whether neuromodulation regulates synapse formation in developing striatum, we examined the number and strength of glutamatergic synapses on SPNs in acute brain slices prepared from P8-13 mice, age-balanced across groups. Comparisons of dendritic spine density and spontaneous miniature excitatory post-synaptic currents (mEPSCs) were made between slices prepared from mice injected 1 hr previously with saline or ligands for GPCRs differentially expressed by dSPNs and iSPNs.

We observed that in acute mouse brain slices prepared 1 hr after systemic injection of the Drd1 agonist SKF 81297 (5 mg/kg), both dendritic spine density and mEPSC frequency in dSPNs were increased compared to saline-injected littermates (*Figure 1C–G*). The frequency of mEPSCs in Drd1 agonist exposed SPNs was 593 ± 168% of that in saline-treated controls (N = 7 and 8 neurons from 4 mice/group, Kruskal-Wallis test p = 0.0063, with p<0.05 for Dunn's multiple comparison post hoc test). mEPSC amplitude was 10.8 ± 1.6 and 12.7 ± 1.1 pA for control and Drd1 agonist groups and was not significantly different (one way ANOVA, p = 0.313). Dendritic spine density was 0.29 ± 0.02 spines/μm and 0.60 ± 0.04 spines/μm for saline control and Drd1 agonist groups respectively (p<0.001 on Dunnett's multiple comparison test after one-way ANOVA; N = 36 dendrites, 12 neurons from 3 mice and 21 dendrites in 7 neurons from 3 mice). Pretreating mice with a PKA antagonist H-89 (5 mg/kg) 30 min prior to agonist injection prevented the effects on mEPSC frequency and spine density (mEPSC frequency, 122 ± 47% of control with Drd1 agonist, N = 6 and 8 neurons from 2 mice/group, p>0.05 for Dunn's multiple comparison post hoc test; spine density, 0.27 ± 0.02 spines/μm vs. 0.29 ± 0.02 spines/μm in control and Drd1 agonist treatment conditions, N = 18 dendrites in 6 neurons from 2 mice, and 24 dendrites in 8 neurons from 2 mice). These results suggest a PKA-dependent increase in the number of AMPA receptor-containing glutamatergic synapses induced by in vivo Drd1 activation. However, Drd1 receptor activation alone is insufficient to increase synapse number, as changes in mEPSC frequency or amplitude were not observed with incubation of acute brain slices in SKF 81297 in vitro (*Figure 1—figure supplement 1*. For mEPSC frequency, 107 ± 23% of ACSF control values with Drd1 stimulation; N = 6 neurons from 2 mice/group, t-test p = 0.83. The amplitude of mEPSCs was 12.3 ± 1.9 and 13.8 ± 2.2 pA for control and Drd1 stimulation groups respectively, t-test p = 0.51). Thus, dSPN excitatory synapse enhancement by a single, acute dose of a Drd1 receptor agonist alone requires intact circuitry, as it cannot be induced in the slice.

In contrast to the effects seen in dSPNs, in vivo administration of SKF 81297 had no effects on excitatory synapses in iSPNs (*Figure 1H*), consistent with the lack of expression of Drd1 receptor on these cells (*Gerfen et al., 1990*) (mEPSC frequency, 73.3 ± 30.1% of control values in presence of Drd1 agonist; N = 6 neurons from 4 mice and 8 neurons from 2 mice). iSPNs express G$\alpha_s$-coupled Adenosine 2a (A2a) receptor, which also enhances PKA activity (*Higley and Sabatini 2010*) – nevertheless, administration of an agonist of A2aRs (CGS 21680, 0.1 mg/kg) had no significant effect on iSPN dendritic spine density or mEPSC rate, within the rapid time-frame of 1 hr (140 ± 63% of control, p>0.05 for control vs. A2aR agonist post-hoc comparison; N = 11 neurons from 3 mice). Similarly, administering an agonist of G$\alpha_i$-coupled Drd2 receptors (quinpirole 0.5 mg/kg) also resulted in no significant differences from controls (132 ± 17.0% of control, p>0.05 for control vs. Drd2 agonist post hoc comparison; N = 8 neurons from 3 mice). However, co-administration of Drd1 and A2a

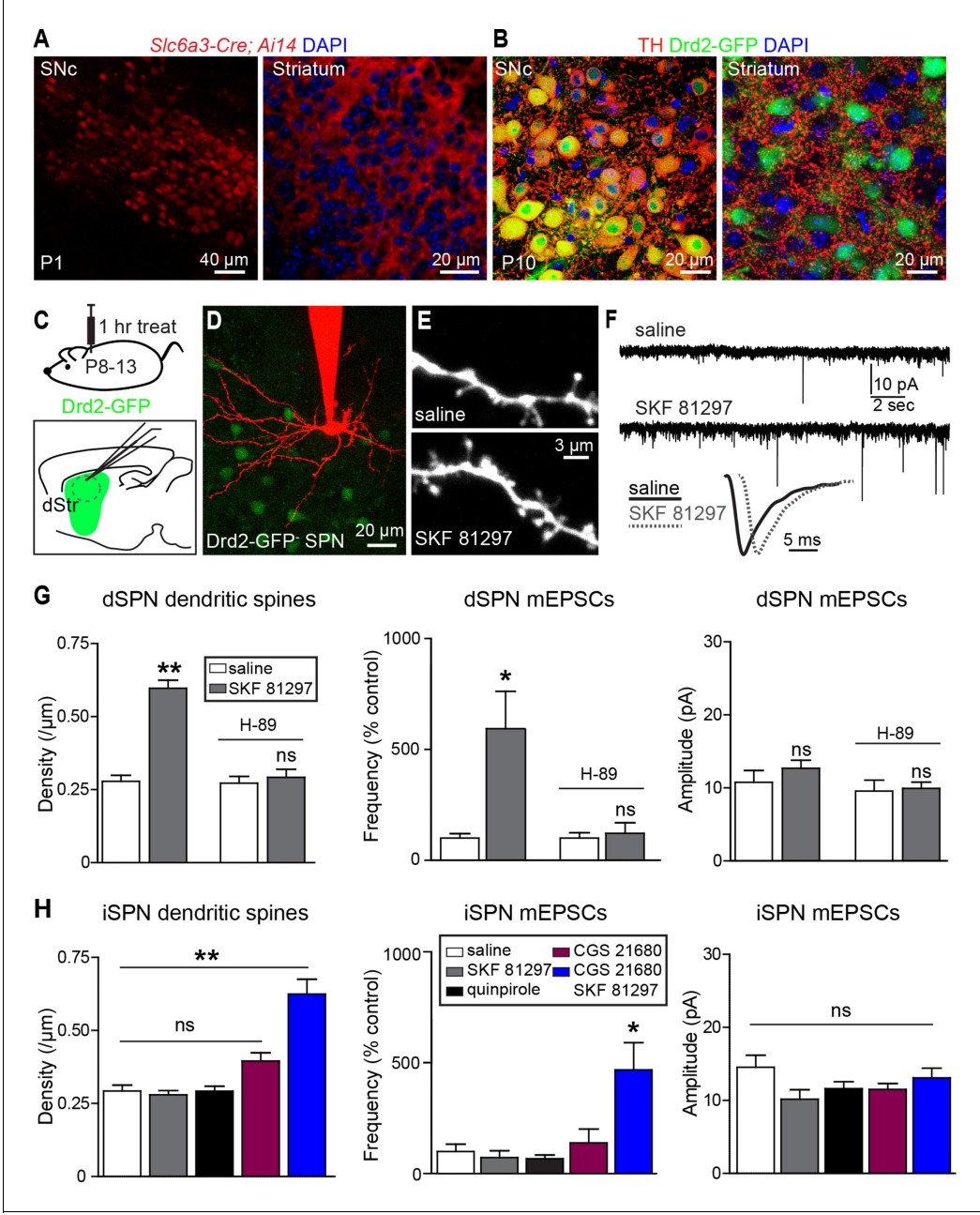

**Figure 1.** Gα$_s$ GPCR stimulation rapidly increases SPN dendritic spine and excitatory synapse number. (**A**) Cre recombinase expression driven by dopamine transporter gene (*Slc6a3*) promoter activates tdTomato expression (red) in a reporter mouse, revealing cells bodies in SNc (*left*) and densely innervating fibers within striatum (*right*) at postnatal day 1 (P1). (**B**) At P10, immunohistochemistry for tyrosine hydroxylase shows expression in SNc cell bodies (*left*) and a punctate pattern in the striatum (*right*). (**C**) Schematic of the experiment and recording preparation. (**D**) 2PLSM image of a Drd2-GFP negative SPN filled with Alexa 594 during whole-cell recording. (**E**) 2PLSM images of dendrites and dendritic spines on dSPNs after saline or SKF 81297 administration in vivo. (**F**) Example mEPSC recordings and individual events demonstrating enhanced mEPSC frequency in dSPNs with Drd1 stimulation. (**G**) Summaries of spine density, mEPSC frequency and amplitude in dSPNs following in vivo administration of saline or the Drd1 agonist SKF 81297 to control animals or those treated with the PKA antagonist H-89. (**H**) Summaries of spine density, mEPSC frequency and amplitude in iSPNs following in vivo administration of saline, the Drd1 agonist SKF 81297, the Drd2 agonist quinpirole, the A2aR agonist CGS 21680, or SKF 81297 and CGS 21680 together.

*Figure 1. continued on next page*

*Figure 1. Continued*

The following figure supplements are available for Figure 1:

**Figure supplement 1.** Drd1 agonist treatment of the acute brain slice fails to enhance excitatory synapse numbers on dSPNs.

**Figure supplement 2.** Acute Drd1 agonist treatment induces locomotion in immature pups.

**Figure supplement 3.** Drd1 agonist treatment repeatedly potentiates locomotion in immature pups.

receptor agonists increased mEPSC frequency and dendritic spine density in iSPNs, with no change in mEPSC amplitude, suggesting an increase in the number of functional excitatory synapses (*Figure 1H*) (mEPSC frequency, 470 ± 124% of control values with CGS 21680/SKF 81297, one-way ANOVA p<0.0008, Bonferroni post hoc p<0.05; N = 6 neurons from 4 mice and 5 neurons from 2 mice for this comparison. Dendritic spine density, 0.29 ± 0.02 spines/μm and 0.63 ± 0.05 spines/μm, for control vs. CGS 21680/SKF 81297, one-way ANOVA p<0.0001, Bonferroni post hoc p<0.0001; 31 dendrites, 9 neurons, 2 mice, and 7 dendrites, 4 neurons in 2 mice, for this comparison).

These results indicate that neither the activation of dopamine receptors, nor of $G\alpha_s$-coupled GPCRs, is sufficient to cell-autonomously enhance glutamatergic synapses on SPNs. Instead, neuro-modulatory GPCR signaling may be a permissive, or facilitating, signal that acts in concert with other stimuli to induce synaptogenesis in vivo. We hypothesized that GPCR activation modulates gluta-mate-induced, activity-dependent synaptogenesis. Since Drd1 activation also increases dSPN excit-ability, which in turn may increase recurrent corticostriatal glutamatergic transmission onto SPNs by altering cortical activity (*Oldenburg and Sabatini, 2015*), then activation of this receptor alone in vivo will (1) activate PKA and (2) enhance glutamate release. This may be sufficient to drive synapse formation. Such stimulatory effects of dopamine on basal ganglia circuitry are consistent with the observation of Drd1 receptor activation-induced locomotion observed in pups, which normally show limited mobility at P8-13 (*Dehorter et al., 2011*; *Wills et al., 2014*) (*Figure 1—figure supplement 2*). This acute locomotion induction has lasting consequences for behavior and brain development, since priming with Drd1 receptor agonist administration at P9 potentiated agonist-evoked locomo-tion at P11 (*Figure 1—figure supplement 3*). In contrast to dSPNs, iSPNs would be expected to have more complex requirements for synaptogenesis, since iSPN activity is negatively coupled to intrastriatal glutamate release in the adult, via regulation of cortical activity (*Oldenburg and Saba-tini, 2015*). We hypothesize that the cooperation of presynaptic glutamate release and postsynaptic PKA signaling explains why coincident activation of Drd1 receptors (to recurrently drive glutamate release) and A2aRs (to enhance PKA signaling) is required in vivo to increase synaptogenesis in iSPNs.

We directly tested this hypothesis ex vivo by examining the ability of exogenous glutamate expo-sure, through focal photolysis of MNI-L-glutamate, to trigger synaptogenesis in SPNs under different states of GPCR activation. Spatio-temporally controlled glutamate release can focally and rapidly induce de novo formation of a synapse-bearing dendritic spine on young cortical pyramidal neurons (*Kwon and Sabatini, 2011*). Cre-dependent AAV was used to induce GFP expression in Cre-expressing dSPNs or iSPNs (*Figure 2A–C*), allowing visualization of the cellular and dendritic mor-phology of sparsely labeled neurons of either pathway. Repeated glutamate uncaging (40 pulses at 2 Hz) with a variety of pulse durations (0.5–4 ms) was applied near a GFP-positive dendrite under conditions that maximally activate NMDA-type glutamate receptors (0 mM added extracellular $Mg^{2+}$). In a subset of cases, this protocol induced de novo growth of a dendritic spine-like protrusion (*Figure 2C*). As a control for non-specific, photodamage-induced changes in tissue fluorescence, in a subset of trials, an interleaved stimulus of identical strength was delivered to a control site >5 μm away from any labeled dendrite (*Figure 2B*, *Figure 2—figure supplement 1*) (0.5 vs. 0 success prob-ability at proximal and distal uncaging locations; 82 trials from 2 mice).

As was previously described for layer 2/3 pyramidal neurons, P8-13 SPNs of both pathways were capable of de novo spinogenesis in response to glutamate uncaging (*Kozorovitskiy et al., 2012*; *Kwon and Sabatini, 2011*). The basal rate of spinogenesis (the probability of induction of a structure shaped like a dendritic spine), as well as its dependence on uncaging pulse width and proximity to

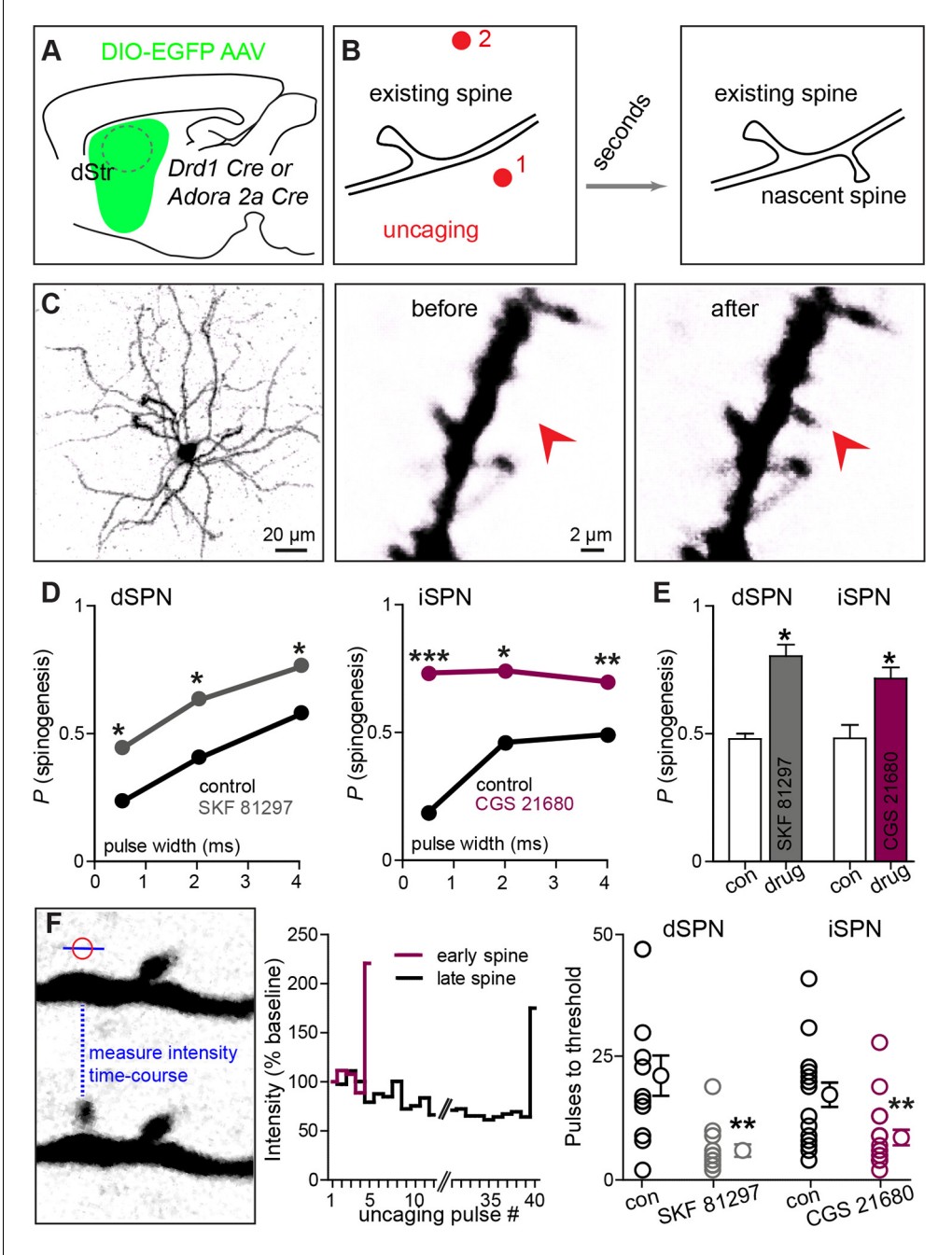

**Figure 2.** Drd1 or A2a receptor activation facilitates spinogenesis on dSPNs and iSPNs, respectively. (**A**) Schematic of experimental setup. AAVs encoding Cre-dependent GFP expression are used with appropriate transgenic mice to induce sparse GFP labeling of dSPNs or iSPNs. (**B**) Schematic illustrating de novo dendritic spine formation triggered by focal 2-photon glutamate uncaging. In a subset of experiments, uncaging was rapidly alternated between two sites, located close to (<2 μm) and far from (>5 μm) the GFP-positive dendrite. De novo spinogenesis occurred in a subset of trials in locations near the dendrite. (**C**) Example of a GFP+ neuron imaged at low magnification (*left*) and higher magnification images of a dendrite before (*middle*) and after (*right*) successful induction of new spine formation (arrow). (**D**) Probability of successful spinogenesis across all induction attempts in dSPNs (*left*) and iSPNs (*right*) as a function of uncaging pulse width in control conditions (black) or in the presence of the indicated Gα$_s$ GPCR agonist (grey, Drd1 agonist SKF 81297; burgundy, A2aR agonist CGS 21680). (**E**) Average probability of spinogenesis for dSPNs and iSPNs analyzed in control conditions or in the presence of the indicated Gα$_s$ GPCR agonist. Error bars indicated SEM of the probability across mice. (**F**) Analysis of spinogenesis

*Figure 2. continued on next page*

*Figure 2. Continued*

kinetics using line scans to measure fluorescence in the emerging spine head (*left*) indicates that although the actual process of spinogenesis is fast (*right*), some spines are generated early (burgundy) and others late (black) during the stimulation protocol. (**G**) Gα$_s$ GPCR agonist treatment reduces the number of uncaging pulses needed for detection of an emergent dendritic spine. Error bars indicate SEM.

The following figure supplements are available for Figure 2:

**Figure supplement 1.** Dual site uncaging de novo spinogenesis induction.

the dendrite were similar for striatal SPNs and previously examined cortical neurons (*Kwon and Sabatini, 2011*). For both dSPNs and iSPNs, the success rate for inductions with stimulation 1–2 μm away from a labeled dendrite was ~20–50%, with longer uncaging pulses associated with a greater probability of spinogenesis (dSPNs, N = 146 and 174 trials in 10 mice for control group and in presence of GPCR agonist, respectively, balanced across conditions; iSPNs, N = 132 and 164 trials in 11 mice). The probability of spinogenesis increased in the presence of SKF 81297 for dSPNs, and CGS 21680 for iSPNs, respectively (the proportion of success trials was 0.23, 0.40, and 0.57 in the control condition, for varied length glutamate uncaging pulses, and 0.44, 0.62, and 0.76 in presence of Drd1 receptor agonist for dSPNs; for iSPNs, the respective proportions were 0.19, 0.46, and 0.49 in the control condition vs. 0.73, 0.74, and 0.70 for iSPNs with A2aR stimulation; two-tailed Z tests for proportions, p = 0.040, 0.015, and 0.018 for dSPNs, and p<0.0001, p = 0.018, and 0.008 for iSPNs). This enhancement was observed for all 3 uncaging pulse-widths tested (*Figure 2D*), and is also evident in across-cell averages within each mouse, for the 4 ms-long uncaging pulse condition (*Figure 2E*; proportion of successful trials, 0.48 vs. 0.80 for dSPNs in control and with Drd1 agonist treatment, and 0.48 vs. 0.72 for iSPNs; dSPNs, N = 3 and 4 mice/group, 68 and 75 trials; iSPNs, N = 3 and 4 mice/group, 61 and 103 trials).

Analysis of the time course of spine formation for successful inductions reveals that Gα$_s$-coupled receptor stimulation decreased the number of glutamate pulses necessary to induce spinogenesis (*Figure 2F*). Whereas spine growth occurs rapidly in all conditions, as observed in pyramidal neurons (*Kwon and Sabatini, 2011*), SKF 81297 and CGS 21680 accelerated the onset of spinogenesis for dSPNs and iSPNs, respectively. We compared the number of photoactivation pulses to threshold, defined as 50% fluorescence increase over baseline in the uncaging location, leading to an appearance of a spine-like extension from the dendrite. At 2 Hz stimulation, for successful trials, threshold pulse number was 21.3 without drug and 6.0 in the presence of Gα$_s$ agonist for dSPNs, and 17.4 vs. 8.7 pulses for iSPNs (Mann-Whitney test, p = 0.0025 and p = 0.0048 for d- and iSPN comparisons, respectively; N = 12 and 13 induction trials for dSPN control vs. in the presence of agonist, with 16 and 17 induction trials for iSPNs, respectively). Given that glutamate was exogenously controlled and Gα$_s$ receptors were stimulated by agonist bath application, these findings suggest the presence of cell-autonomous mechanisms for neuromodulatory regulation of glutamate-dependent spinogenesis.

If a similar interaction of neuromodulation and glutamate release controls synapse formation in vivo, it should be possible to bias rapidly developing networks towards increased synaptogenesis by briefly enhancing either PKA signaling or glutamate release. To test this prediction for PKA signaling, we used a pharmacogenetic approach based on Rs1, a modified Gα$_s$-coupled GPCR sensitive to a small, exogenous, blood-brain-barrier permeable ligand (*Srinivasan et al., 2003*). The efficacy of this GPCR coupling to Gα$_s$ was examined in engineered HEK293 cells, in which GPCR coupling to Gα$_s$ elevates cyclic AMP and activates a transcriptional cAMP response element, leading to the production of secreted alkaline phosphatase (SEAP) (*Liberles and Buck, 2006*). Transfection of HEK293 cells with Rs1, followed by incubation in the agonist RS 67333 and chemi-luminescent analysis of SEAP levels revealed a sub-micromolar EC$_{50}$ of Rs1 activation (*Figure 3A*).

We tested the sufficiency of Gα$_s$-coupled GPCR activation in SPNs for enhancing corticostriatal transmission, using a combination of pharmacogenetic activation in vivo and optogenetic analysis of evoked corticostriatal transmission ex vivo (*Figure 3B–G*). We compared the magnitude of EPSCs evoked by channelrhodopsin (ChR2)-mediated activation of Rbp4-Cre expressing neurons, which include a dense corticostriatal projection, in SPNs in acute brain slices from animals with and without

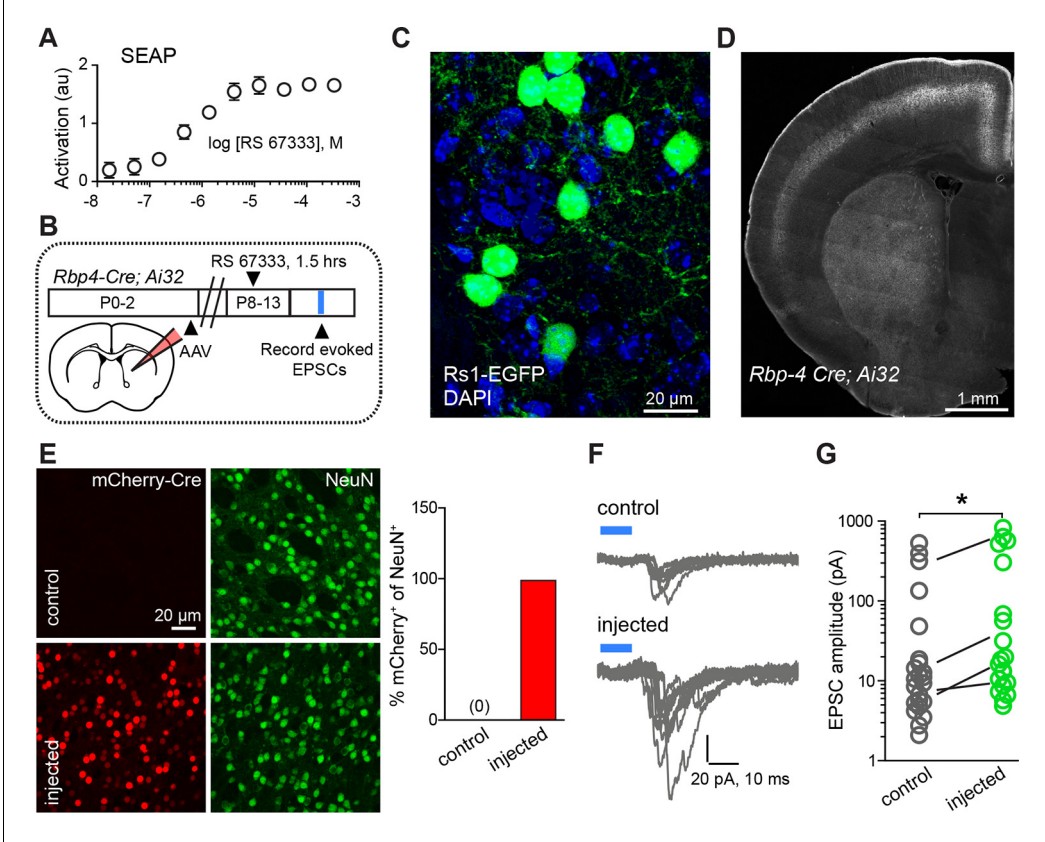

**Figure 3.** Chemogenetic Gα$_s$ GPCR activation is sufficient to rapidly enhance corticostriatal innervation. (**A**) Secreted alkaline phosphatase (SEAP)-based in vitro test of the efficacy of PKA activation following activation of the Gα$_s$-targeted RASSL with RS 67333. (**B**) Experimental design showing unilateral neonatal virus injection of DIO-Rs1-EGFP AAV and mCherry-Cre AAV into striatum of *Rbp4-Cre; Ai32* mice. At P8-13, pups were systemically injected with RS 67333 and acute slices were prepared 1.5 hr after the injection. Light-evoked EPSCs were recorded in SPNs and compared across injected and uninjected hemispheres. (**C**) Confocal image showing striatal expression of a Cre-dependent Gα$_s$ targeted Rs1-EGFP AAV in a Cre-expressing mouse. (**D**) YFP fluorescence image illustrating the expression ChR2-YFP in layer 5 pyramidal neurons in coronal brain section of an *Rbp4-Cre; Ai32* mouse. (**E**) Images mCherry-Cre (*left*, red) and NeuN (*middle*, green) expression in control (*top*) or Cre-mCherry encoding AAV injected (*bottom*) striatum. The proportion of striatal neurons identified immunohistochemically by NeuN expression that co-express Cre in infected or uninfected hemispheres. (**F**) Examples of light-evoked EPSCs from SPNs in control and virally injected hemisphere of the same mouse. Ten overlaid consecutive acquisition traces are shown for each neuron. (**G**) Summary data showing larger amplitudes of ChR2-evoked EPSCs for SPNs from the injected side of the brain. Circles are average responses of single neurons, whereas lines reflect individual mouse averages.

The following figure supplements are available for Figure 3:

**Figure supplement 1.** Age dependence of optogenetically evoked corticostriatal responses.

Rs1 expression. *Rbp4-Cre; Ai32* mice were unilaterally injected in the striatum with a Cre-expressing AAV and Cre-dependent AAV encoding Rs1 (*Hsiao et al., 2008*) (*Figure 3C–D*). This co-injection resulted in broad expression in dorsal striatum SPNs of the injected hemisphere in both dSPNs and iSPNs (*Figure 3E*) (0% vs. 98% co-labeling, N = 750 and 728 neurons in control and injected hemispheres, from 3 mice). P8-13 infected mice were injected with the Rs1 ligand RS 67333 (3 mg/kg) and acute brain slices were prepared 1.5 hr later for analysis. Whole-cell voltage clamp recordings were used to monitor pharmacologically isolated, AMPAR-mediated blue light-evoked corticostriatal EPSCs, which were recorded from SPNs in dorsal striatum, located either in the control or in the infected hemisphere. Although this strategy results in ChR2 expression in SPNs, the brief time of

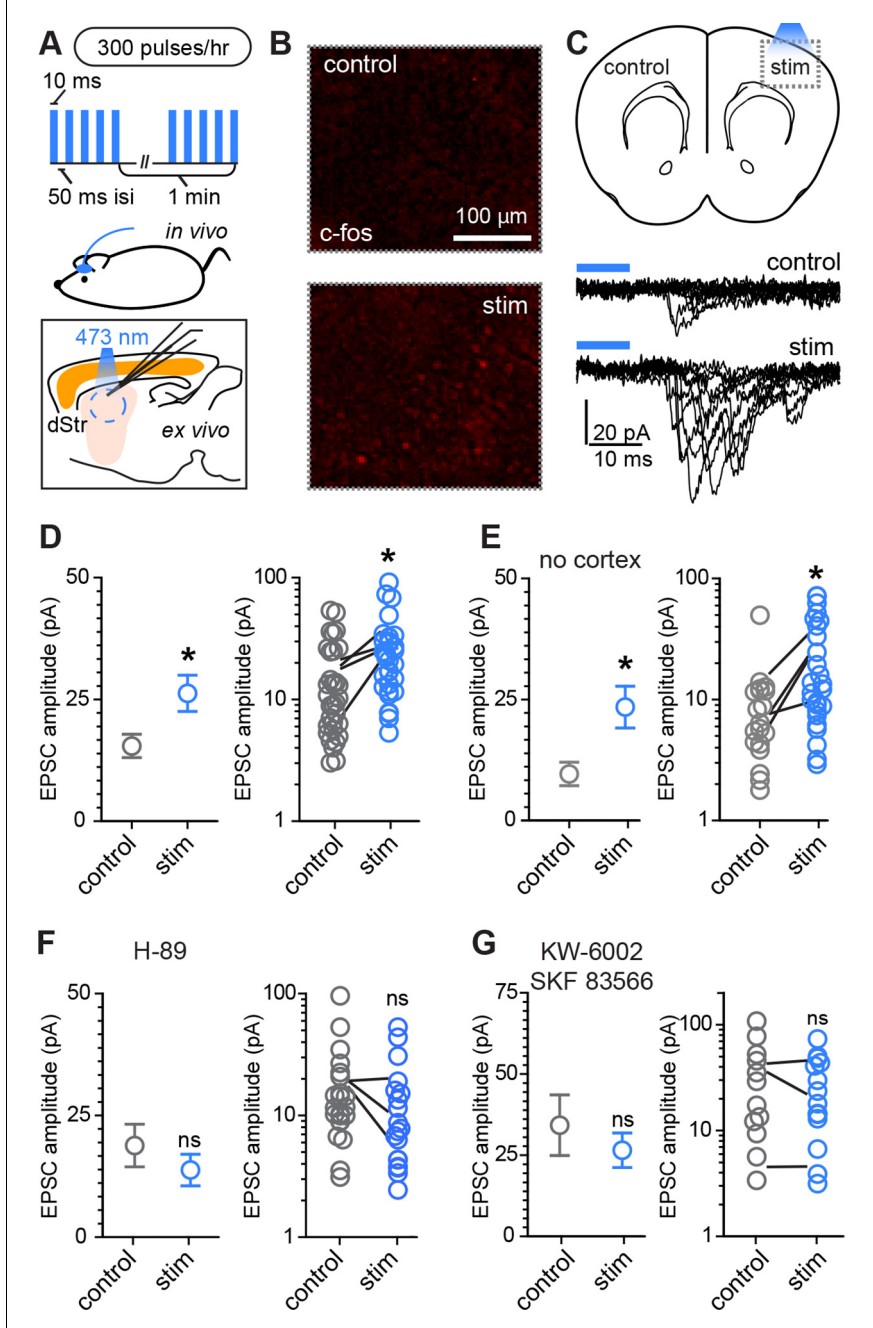

**Figure 4.** Rapid activity-dependent in vivo corticostriatal plasticity depends on PKA and G$\alpha_s$-coupled receptor activation. (**A**) Experimental schematic. A mouse pup expressing ChR2-YFP in corticostriatal fibers (*Rbp4-Cre; Ai32*) receives 300 light pulses over the course of 1 hr via an extracranially mounted LED located over somatosensory and motor cortices. Immediately after in vivo optogenetic stimulation, acute slices are prepared and ChR2-mediated EPSCs in SPNs are recorded from stimulated and control hemispheres. (**B**) Immunohistochemical labeling of immediate early gene c-fos product shows increased labeling on the optogenetically stimulated side. (**C**) Schematic of the experiment and example traces showing ten sequentially evoked EPSCs from example SPNs of the control and stimulated hemisphere in one mouse. (**D**) Average amplitudes of optogenetically evoked EPSCs showing successful induction of corticostriatal plasticity under baseline conditions. Group averages are shown on the left (error bars reflect SEM) and single neuron (circles) as well as individual mouse (lines) averages are on the right. (**E**) As in **D**, repeated in in striatum-only slices, prepared by removing cortex before recording from SPNs. (**F**, **G**) As in **D**, showing in vivo pre-administration of PKA blocker H-89 30 min before optogenetic stimulation (**F**) or

*Figure 4. continued on next page*

*Figure 4. Continued*

antagonism of Drd1 and A2a receptors with SKF 83566 and KW-6002 (**G**) prevent optogenetically induced corticostriatal plasticity.

The following figure supplements are available for Figure 4:

**Figure supplement 1.** Light-evoked response properties in SPNs of *Rbp4-Cre; Ai32* mice.

---

expression from a single floxed ChR2 allele yielded negligible direct ChR2-evoked currents in SPNs under whole-cell voltage clamp configuration (*e.g.*, *Figure 3F*). With this strategy, both the control and infected striatum were exposed to RS 67333 and comparisons were made across SPNs from control and infected hemispheres in the same animal. In the control condition, there was a strong age dependence of peak amplitude of EPSCs, ranging from an average of 10 pA to an average of 196 pA per age group, in a cohort of P8-P13/14 mice (*Figure 3—figure supplement 1*; light evoked EPSC amplitude across age, N = 14 neurons from 3 mice, 30 neurons from 6 mice and 14 neurons from 3 mice/group. Kruskal-Wallis test p<0.0001, for main effect and Dunn's post hoc tests). Consistent with our hypothesis, light-evoked EPSC amplitude was increased in virally transduced SPNs, compared to control hemisphere SPNs (*Figure 3G*; light-evoked EPSC amplitude, 68.6 ± 29.9 pA control vs. 150.3 ± 55.8 pA injected hemisphere; N = 21 and 23 neurons from 4 mice/group, t-test p = 0.039). Thus, in vivo transient activation of a $G\alpha_s$-coupled GPCR in SPNs is sufficient to rapidly enhance corticostriatal connectivity, in the presence of otherwise normal corticostriatal transmission. The observed sufficiency of pharmacogenetic PKA activation to enhance corticostriatal connectivity for iSPNs (in contrast to stimulating iSPNs with an A2aR agonist, as in *Figure 1H*) stems from the likelihood that the concurrent pharmacogenetic activation of $G\alpha_s$ cascades in dSPNs enhances corticostriatal glutamatergic drive via recurrent connections and regulating cortical activity (*Oldenburg and Sabatini, 2015*).

Conversely, to examine whether increased glutamate release in vivo is able to enhance corticostriatal transmission during this developmental period, a small blue light LED was mounted over somatosensory cortex on one hemisphere and used to activate ChR2-expressing Rbp4-Cre positive neurons in cortex (*Figure 4*). Mice were awake and freely moving within an enclosure, when a mild stimulus was delivered, consisting of 300 pulses of light over the course of 1 hr, given in bursts of 5 pulses at 20 Hz every minute (*Figure 4A*). This treatment induced moderate c-fos expression in the stimulated cortical region (*Figure 4B*). Acute slices were prepared immediately after the stimulation protocol and AMPAR-mediated light-evoked EPSC were recorded from dorsal striatum SPNs, located in the control or stimulated hemisphere. Both EPSC peak amplitude and charge transfer were increased on the stimulated side (*Figure 4D*, *Figure 4—figure supplement 1*) (Light-evoked EPSC amplitude, 30.5 ± 4.8 pA vs. 50.4 ± 6.6 pA for control and stimulated hemisphere comparison; charge transfer, 0.32 ± 0.14 pC and 0.89 ± 0.21 pC; N = 29–33 neurons/group, 4 mice/group). Paired analysis of EPSC amplitude within subjects across the two hemispheres is consistent with individual cell data (*Figure 4D*), confirming the successful induction of corticostriatal rewiring following cortical optogenetic stimulation (EPSC amplitude, control 16.2 ± 3.2 vs. 27.3 ± 3.2 pA in the stimulated hemisphere, paired t-test p = 0.026, 4 mice/group).

Because stimulating layer 5 pyramidal neurons may alter intra-cortical connectivity, which could indirectly account for the observed differences in SPN responses, we tested whether this form of plasticity is striatally expressed by trimming off cortex from acute coronal brain slices with a fine scalpel prior to recording (*Figure 4E*, *Figure 4—figure supplement 1*). Both peak EPSC amplitude and charge transfer were increased on the stimulated side (light-evoked EPSC amplitude, 9.6 ± 2.4 pA vs. 23.4 ± 4.3 pA for control vs. stimulated hemisphere; charge transfer, 0.64 ± 0.14 and 1.2 ± 0.26 pC for the same comparison; N = 19 and 26 neurons from 4 mice/group). These data demonstrate that a major component of the functional reorganization induced through brief in vivo stimulation of corticostriatal afferents in young mice is locally expressed in the striatum. We further probed the pharmacological dependence of this form of plasticity and found that it was blocked by either pre-administration of PKA inhibitor H-89 (5 mg/kg), or by a combination of Drd1 and A2aR blockers SKF 83566 (0.5 mg/kg) and istradefylline/KW-6002 (2.5 mg/kg), administered 30 min before optogenetic stimulation (*Figure 4F–G*). Light-evoked EPSC amplitude, in presence of H-89, was 18.8 ± 4.4

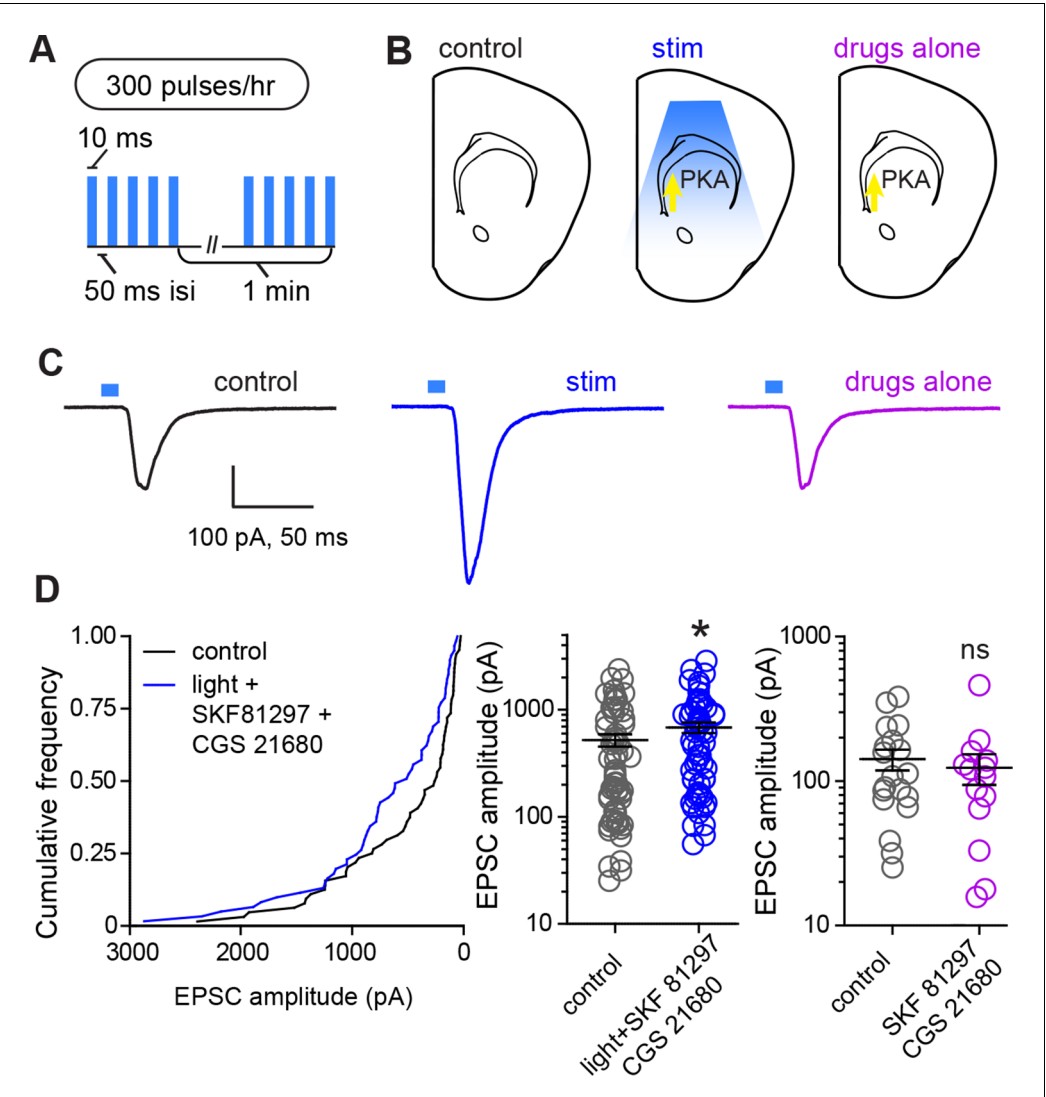

**Figure 5.** Glutamate release and G$\alpha_s$-coupled receptor activation are sufficient to rapidly enhance corticostriatal connectivity in brain slices. (**A**) Experimental schematic for optogenetic stimulation parameters. (**B**) Graphic representation of stimulation conditions in the acute slice. (**C**) Example traces showing average evoked EPSCs for representative neurons in control, stimulation (light+G$\alpha_s$-coupled receptor agonists) and drugs only conditions. (**D**) Average amplitudes of optogenetically evoked EPSCs. *Left*, cumulative distribution of responses in control and stimulated groups. Group averages are shown on the middle and right graphs. Single neurons are represented by circles, error bars reflect SEM.

pA vs. 13.8 ± 3.3 pA in neurons from control and stimulated hemispheres (t-test p = 0.376, N = 22 and 19 neurons from 3 mice/group). In presence of Drd1 and A2aR blockers, it was 34.2 ± 9.3 pA and 26.5 ± 5.3 pA, respectively (t-test p = 0.458, N = 12 and 15 neurons from 3 mice/group). Thus, SPN changes induced by glutamatergic stimulation require PKA signaling, which in turn is activated via Drd1 and A2aR for dSPNs and iSPNs, respectively. Increasing either pre-synaptic glutamatergic drive or enhancing striatal PKA signaling is sufficient to rapidly alter SPN synaptogenesis, supporting the idea of cooperative, neuromodulation-dependent circuit remodeling during striatal development.

In order to directly evaluate whether PKA activation and glutamate release within striatum are sufficient for strengthening corticostriatal synapses, we performed ex vivo plasticity induction using optogenetic stimulation of corticostriatal afferents in the presence of drugs activating PKA in SPNs

(*Figure 5*). We found that the same optogenetic stimulation parameters that evoked PKA-dependent enhancements in evoked responses in vivo were sufficient to produce a moderate enhancement in light-evoked corticostriatal responses in vitro. In experiments using within-mouse controls, the amplitude of evoked responses was 528.8 ± 69 pA and 694.4 ± 79 pA, for neurons in control and stimulated slices respectively (*Figure 5B–C*) (Mann-Whitney test, p = 0.037, N = 64 and 61 neurons from 4 mice). In contrast, incubation in PKA agonists alone was insufficient to alter evoked response amplitude (Mann-Whitney test, p = 0.6, N = 19 and 14 neurons). These data directly confirm the existence of a mechanism for cooperative, neuromodulation-dependent circuit remodeling during striatal development.

## Discussion

We examined the hypothesis that neuromodulation interacts with glutamate-dependent circuit wiring in the developing striatum. Instead of a privileged action of dopamine in striatal synapse development, we discovered a mechanism of regulated synaptogenesis requiring cooperative action of glutamatergic drive and neuromodulation. Our data reveal that neuromodulation, through the activity of PKA, enhances the probability of dendritic spine formation and lowers the amount of glutamate necessary to trigger spinogenesis in the developing striatum.

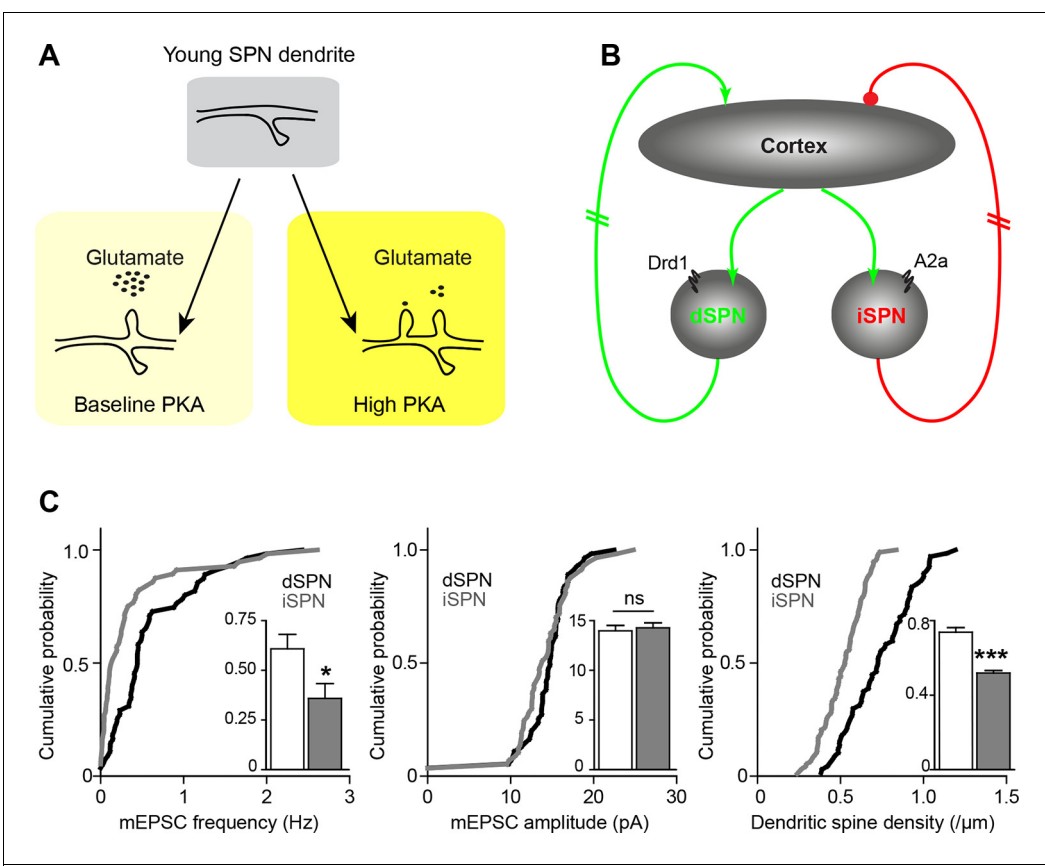

**Figure 6.** Pathway bias and neuromodulatory control of SPN excitatory synaptogenesis during development. (A) Schematic summarizing the key result of this study. Neuromodulation, acting via PKA, regulates glutamate-dependent developmental synaptogenesis. (B) These results fit into the larger framework of the basal ganglia model, which indicates that through polysynaptic recurrent loops involving the neocortex, dSPN activity is positively coupled to striatal glutamate release onto SPNs of both pathways, whereas indirect pathway activity provides negative feedback. (C) Together the two models can account for a transient developmental bias towards the earlier innervation of dSPNs over iSPNs. During development, dSPNs have greater dendritic spine density and miniature EPSC frequency but not amplitude, compared to iSPNs. These graphs represent re-analyses of control data from a previously published study (*Kozorovitskiy et al., 2012*).

These findings must be integrated into the context of basal ganglia circuitry, where inhibitory outputs of the direct and indirect pathways form functionally complementary loops that provide polysynaptic recurrent positive and negative feedback for excitatory striatal synaptogenesis (*Kozorovitskiy et al., 2012*). Because dopamine tends to increase dSPN PKA activity and decrease iSPN PKA activity (models, *Figure 6A–B*), iSPN spinogenesis may be effectively yoked to dSPN development, as described below. Dopamine is abundant in the developing striatum, and it enhances PKA activity in dSPNs. This allows dSPNs to strongly respond to glutamate release from the first invading axons, becoming integrated into early basal ganglia circuits. On the other hand, iSPN activity and PKA levels are decreased by dopamine (*Hernandez-Lopez et al., 2000*) and their excitatory wiring may rely either on the positive feedback through direct pathway (*Kozorovitskiy et al., 2012*) to increase glutamate release in the striatum, or on striatal adenosine levels that may be driven by local activity. This scenario predicts a transient developmental bias in pathway innervation towards dSPNs, which we have previously observed in the frequency of mEPSCs, as well as in dendritic spine density at P14-15 (*Figure 6C*) (mEPSC frequency, $0.61 \pm 0.1$ and $0.35 \pm 0.1$ Hz for d- and iSPNs, t-test p = 0.019; mEPSC amplitude, $14.3 \pm 0.5$ and $14.0 \pm 0.5$ pA, p = 0.679; dendritic spine density, $0.74 \pm 0.025$ and $0.52 \pm 0.013$ spines/μm, p<0.0001; N = 55 neurons from 12 mice and 56 neurons from 11 mice for electrophysiology; a distributed subset of recorded neurons was imaged for dendritic spine density analyses, 67 dSPN dendrites from 22 neurons and 93 iSPN dendrites from 31 neurons). Thus, the palette of neuromodulators and their postsynaptic coupling in a given brain region or developmental time window orchestrate the integration of each neuronal population into functional circuitry.

The observation that two types of $G\alpha_s$-dependent modulation appear to regulate synaptogenesis in d- and iSPNs raises the possibility that PKA activity, via $G\alpha_s$ GPCRs, generally serves to potentiate activity-dependent synaptogenesis during the time of rapid synapse production. The array of $G\alpha_s$ GPCRs expressed in the central nervous system is substantial and includes abundant receptors such as $5-HT_4$ and $5-HT_7$ serotonin receptors (*Giulietti et al., 2014*), as well beta-adrenergic (*Nomura et al., 2014*; *Arriza et al., 1992*) and corticotropin-releasing factor receptors (*Fu and Neugebauer, 2008*; *Riegel and Williams, 2008*). A general role for neuromodulators acting on $G\alpha_s$ GPCRs during neural circuit development would transform our understanding of core principles in developmental neurobiology, greatly expanding the list therapeutic targets for neurodevelopmental and neurodegenerative disorders.

## Materials and methods

### Mouse strains and genotyping

Animals were handled according to protocols approved by the Harvard Standing Committee on Animal Care, in accordance with the guidelines described in the US National Institutes of Health Guide for the Care and Use of Laboratory Animals. In order to identify direct- and indirect-pathways SPNs in experiments described in *Figure 1* and related supporting data, we used *Drd2-EGFP* transgenic mice (GENSAT, #RP23-161H15). For de novo spinogenesis experiments, *Drd1a-Cre* (GENSAT, founder line EY262) (*Gong et al., 2003*) and *Adora2A-Cre* (*Gong et al., 2007*) (GENSAT, founder line KG139) mouse pups were generated by crossing a parent carrying a single Cre-positive allele to C57BL6 wild-type mice. Genotyping primers and PCR protocols have been previously described (*Kozorovitskiy et al., 2012*). Experiments were also performed on mice that carried the *Rbp4-Cre* transgene (GENSAT, #RP24-285K21) crossed to floxed TdTomato reporter mice (Ai14, Jackson Lab, #007914) (*Madisen et al., 2010*) or floxed ChR2 (H134R) mice (Ai32, Jackson Lab, Bar Harbor, ME, #012569). For detecting dopamine transporter (DAT) activity at a young age, knock-in mice expressing Cre recombinase under DAT promoter, *Slc6a3-ires-Cre* (Jackson Lab, # 006660) (*Bäckman et al., 2006*), were crossed to Ai14 mice. To evaluate locomotor activity in mouse pups, P8-13 mice were videotaped from above in square or rectangular enclosures, for 30–45 min, 15 min following a single or repeat i.p. administration of saline or receptor agonists (doses are described below). Euclidean distance traveled was calculated from body center positions in x,y space over time.

## Viruses and intracranial injections

Conditional expression of EGFP in Cre-containing neurons was achieved using recombinant adeno-associated viruses (AAVs) encoding a double-floxed inverted open reading frame (DIO) of EGFP, as described previously (*Kozorovitskiy et al., 2012*). Double-floxed Rs1-EGFP AAV (*Srinivasan et al., 2003*) was co-delivered with an AAV expressing Cre fused to mCherry, in order to drive expression in most striatal neurons. Vector DNA was amplified in recombination deficient bacteria (Stbl3, Invitrogen, Thermo Fisher Scientific, Waltham, MA) and packaged in the vector core of University of North Carolina. P0-3 day old mice were cryoanesthetized, received ketoprofen for analgesia, and were placed on a cooling pad. Virus was delivered at a rate of 100 nl/minute using an UltraMicro-Pump (World Precision Instruments, Sarasota, FL). Dorsal striatum was targeted by directing the needle approximately 1 mm anterior to midpoint between ear and eye, 1.5 mm from midline and 1.8 mm ventral to brain surface. Coordinates were slightly adjusted based on pup age and size. After 200–300 nl injections and wound closure, mice were warmed on a heating pad and returned to home cages.

## Acute slice preparation and electrophysiology

Coronal striatal slices were prepared as described previously (*Kozorovitskiy et al., 2012*). Animals were deeply anesthetized by inhalation of isoflurane. Cerebral hemispheres were removed and placed in cold choline-based artificial cerebrospinal fluid (choline-ACSF) containing 25 mM NaHCO3, 1.25 mM NaH2PO4, 2.5 mM KCl, 7 mM MgCl2, 25 mM glucose, 1 mM CaCl2, 110 mM choline chloride, 11.60 mM ascorbic acid, and 3.10 mM pyruvic acid, and equilibrated with 95%$O_2$/5%$CO_2$. Tissue was blocked and transferred to a slicing chamber containing choline-ACSF. Two hundred and seventy-five or three hundred μm-thick slices were cut on a Leica VT1000s (Leica Instruments, Buffalo Grove, IL) and transferred into a holding chamber with ACSF containing (in mM) 127 NaCl, 2.5 KCl, 25 NaHCO3, 1.25 NaH2PO4, 2.0 CaCl2, +/-1.0 MgCl2, and 25 glucose, equilibrated with 95% $O_2$/5% $CO_2$. Slices were incubated at room temperature, or at 34°C, for 20–30 min prior to imaging or electrophysiological recording, respectively. In a subset of experiments, cortex was removed from the striatum in coronal sections using a microsurgical knife (EMS, Hatfield, PA).

Whole-cell recordings were obtained from striatal SPNs visualized under infrared differential interference contrast (IR-DIC) using patch pipettes with pipette resistance of ~2–4 MΩ. Epifluorescent illumination enabled identification of infected or GFP-positive SPNs. For miniature excitatory postsynaptic current (mEPSC) recordings and for light-evoked responses, the internal solution consisted of 120 mM CsMeSO4, 15 mM CsCl, 8 mM NaCl, 10 mM TEACl, 10 mM HEPES, 2 mM QX314, 4 mM MgATP, 0.3 mM NaGTP (pH 7.4). Alexa Fluor 594 (10–20 μM) was added to internal to visualize cell morphology and confirm cell identity as SPN. Recordings were made from cells held at -70 mV using Axoclamp 200B or 700B amplifiers (Axon Instruments, Union City, CA) at room temperature. Data were filtered at 3 kHz and sampled at 10 kHz. Series resistance, measured with a 5 mV hyperpolarizing pulse in voltage clamp, was under 20 MΩ and was left uncompensated. All responses were examined for light-evoked EPSC analysis. Miniature EPSC amplitude cut-off for analysis was 6 pA and cells that had no mEPSCs during the recording were included. Neurons with high root-mean-square current noise values (~$I_{RMS}$>2.5 pA) were excluded, regardless of series resistance and holding current. Several minutes following breaking in, ~5 min of continuously acquired 30 second long sweeps were collected and analyzed offline per neuron. The amplitude of mEPSCs amplitude reflects absolute value of inward currents for neurons held at -70 mV in all figures. For normalized mEPSC frequency calculation, the mean of the entire control group was considered 100%. Ages of mice were balanced across drug treatment groups and were always P8-13, except for P14-15 mEPSC and dendritic spine data reanalyzed from a previously published paper in *Figure 6C* (*Kozorovitskiy et al., 2012*), as well as light-evoked EPSC data in *Figure 3—figure supplement 1*, which includes a P13-14 group of mice after eye-opening.

## Optogenetic stimulation in vivo and ex vivo

A bare Cree XPE emitter blue LED was used for in vivo unilateral optogenetic stimulation. P8-14 mice were anesthetized and received analgesia. The emitter surface was flattened and attached directly to the skull over somatosensory cortex on one side. Skin was closed over the emitter, and upon recovery from anesthesia, awake mice were transferred into the stimulation chamber where

they could move freely. Three hundred 10 ms-long pulses (~15 mW outside the cranium) were delivered through the skull. Pulses were delivered in a train of 5 stimuli at 20 Hz, once per minute. Brain slices were prepared immediately after completion of stimulation. For ex vivo whole cell recording of light-evoked EPSCs in SPNs, light from a 473 nm laser (Optoengine, Midvale, UT) was focused on the back aperture of the microscope objective to produce wide-field illumination. Brief pulses of light (10 ms duration, 2–3 mW under the objective) were delivered at the recording site at 30 second intervals. Epifluorescence illumination was used sparingly to minimize ChR2 activation prior to recording and was never used with a GFP filter cube.

For ex vivo optogenetic stimulation of acute brain slices we used a custom built chamber with blue (470 nm) LEDs set perpendicular to the plane of slices. Slices were incubated in ACSF with 5 µM CGP 55845, 10 µM SR 95531, 10 µM SKF 81297, 10 µM CGS 21680 at room temperature and stimulated with 10 mW.mm$^{-2}$ light pulses (5 ms) for one hour. Light pulses were triggered with a Master-8 stimulator and the pulse pattern used for stimulation was 5 pulses at 20 Hz every minute.

## Two-photon imaging with two-photon glutamate uncaging

Dendritic imaging and uncaging of MNI-glutamate for spinogenesis induction were accomplished on a custom-built microscope combining two-photon laser-scanning microscopy (2PLSM) and two-photon laser photoactivation (2PLP), as previously described (Kwon and Sabatini, 2011). Two mode-locked Ti:Sapphire lasers (Chameleon lasers, Coherent, Santa Clara, CA ) were tuned to 840 and 725 nm for exciting GFP or Alexa 594 fluorescence and uncaging, respectively. The intensity of each laser was independently controlled by Pockels cells (Conoptics, Danbury, CT). A modified version of ScanImage software was used for data acquisition (Pologruto et al., 2003) (https://github.com/bernardo-sabatinilab). For glutamate uncaging, 2.5 mM MNI-caged-L-glutamate was perfused into the slice chamber, and 10–15 mW of 725 nm light at the specimen plane (60X objective, Olympus, Waltham, MA) was used to focally release the caging group. Secondary and tertiary dendritic branches were selected for dendritic imaging and spinogenesis induction. MNI-glutamate was uncaged near the dendrite (~0.5 or >5 µm away from the edge) at 2 Hz using up to forty 0.5, 2 or 4 ms-long pulses. Images were continually acquired during the induction protocol at 1 Hz, and uncaging was stopped if an apparent spinehead was visible before 40 uncaging pulses were delivered. For analysis of the spinogenesis timecourse and the 2-site uncaging comparisons, the intensity of pixels along a line crossing the middle of the spinehead (or uncaging spot if no detectable spine appeared) was measured across consecutive images (ImageJ, NIH) and expressed as percentage of maximum fluorescence reached in the uncaging locus 1–2 min following induction. For dendritic spine density analyses, stacks through secondary and tertiary dendrites were acquired, coded and analyzed in MATLAB (MathWorks, Natick, MA) and ImageJ.

## Pharmacology

Pharmacological agents were acquired from Tocris (Bristol, UK) or Sigma-Aldrich (St. Louis, MO). For mEPSC recordings, ACSF contained 1 µM TTX, 50 µM SR 95531/Gabazine, 10 µM CPP, and 10 µM Scopolamine hydrobromide. For light evoked EPSC recordings, SR 95531 and CPP were used at the same concentrations as for mEPSC recordings. Acute slices were treated with 10 µM SKF 81297 or CGS 21680. In vivo treatment of mice included intraperitoneal or subcutaneous injections of SKF 81297 (2.5–5 mg/kg), CGS 21680 (0.1 mg/kg), quinpirole (0.5 mg/kg), H-89 (5 mg/kg), istradefylline/KW-6002 (2.5 mg/kg), SKF 83566 (0.5 mg/kg), and RS 67333 (3 mg/kg).

## Fixed tissue preparation, immunohistochemistry and imaging

Mice were transcardially perfused with 4% paraformaldehyde and brains were post-fixed for 1–5 days, prior to sectioning at 40–50 µm on a Vibratome. No immunoenhancement was used to increase the signal of virally transduced fluorescent proteins. For tyrosine hydroxylase (TH), NeuN and c-fos immunofluorescence, 1:4 series of coronal sections through the striatum or SNc were incubated overnight at 4°C with rabbit anti-TH, mouse anti-NeuN, or rabbit anti-c-fos antibody in TBS with 0.5% Tween-20 (1:500–1000, all from EMD Millipore, Billerica, MA). The following day tissue was rinsed in TBS, reacted with goat anti-rabbit or anti-mouse Alexa Fluor 594 (1:500, Molecular Probes, Thermo Scientific) for 1 hr at RT, rinsed, mounted onto superfrost slides, dried and cover-slipped under ProLong antifade reagent with DAPI (Molecular Probes). Images were acquired with a

Zeiss LSM 510 META or a Leica SP8 confocal microscope (Harvard NeuroDiscovery Center). For quantitative estimates of mCherry-Cre and NeuN overlap, slides were coded for confocal microscopy and data analysis. Z-stacks matching the dorsal targeting of electrophysiological recordings and viral injections were selected for analysis. Two-dimensional 1 μm-thick optical sections were analyzed in ImageJ (FIJI) (*Schindelin et al., 2012*). The percentage of Cre[+] and Cre[-] among NeuN[+] neurons was quantified in both hemispheres.

## Cell culture and SEAP assay

HEK293 cells were grown in DMEM containing 5% FBS and 500 μg/ml G-418 (all Invitrogen) and maintained at 37°C in an atmosphere of 5%$CO_2$. Cells were plated in 96-well plates at 30,000 cells/well and transfected with the SEAP reporter plasmid using Lipofectamine® and PLUS® reagent (Invitrogen). The transfection media was replaced with ligand-containing DMEM (200 μl/well) and cells were incubated in the dark for 24 hrs at 37°C with 5%$CO_2$. After transferring 100 μl aliquots from each well to a fresh 96-well plate, 100 μl of an aqueous buffer containing 2 M diethanolamine bicarbonate and 1.2 mM methylumbelliferone phosphate, pH 10.0, was added to each well. Plates were imaged on a Perkin Elmer Envision plate reader using optical settings for methylumbelliferone fluorescence (Perkin Elmer, Waltham, MA). Data reflect an average of 6 condition replicates, run in parallel during two independent runs of 6 replicates each, done on different days.

## Data analysis

Offline analysis of electrophysiology data was performed using Igor Pro (Wavemetrics, Portland, OR) and MATLAB. Statistical analyses were done using GraphPad PRIZM 5 software (GraphPad, LaJolla, CA). All data except for probabilities are expressed as mean +SEM. Probabilities are expressed as aggregate probabilities across experiments of the same type. For two-group comparisons, statistical significance was determined by two-tailed Student's t-tests or two-sample Z test for proportions. For multiple group comparisons, one-way analysis of variance (ANOVA) tests were used for normally distributed data, followed by post hoc analyses. For non-normally distributed data, non-parametric tests for the appropriate group numbers were used, such as Mann-Whitney and Kruskal-Wallis. $p < 0.05$ was considered statistically significant.

## Acknowledgements

The authors thank Rachel Pemberton for genotyping, Matt Banghart and Ruchir Shah for help with the SEAP assay, Lai Ding (Harvard NeuroDiscovery Center) for ImageJ macros, and all members of the Sabatini laboratory for helpful discussions. YK was supported by Leonard and Isabelle Goldenson Research Fellowship, the Nancy Lurie Marks Family Foundation, and William N. and Bernice E. Bumpus Foundation Innovation Award. RP was supported by the Alice and Joseph Brooks fellowship and the Nancy Lurie Marks Family Foundation. This work was supported by NS046579 and Howard Hughes Medical Institute (BLS).

## Additional information

### Funding

| Funder | Grant reference number | Author |
|---|---|---|
| National Institute of Neurological Disorders and Stroke | | Arpiar Saunders<br>Bernardo L Sabatini |
| Howard Hughes Medical Institute | | Bernardo L Sabatini |
| Nancy Lurie Marks Family Foundation | | Yevgenia Kozorovitskiy<br>Rui Peixoto |
| William N. and Bernice E. Bumpus | Innovation Award | Yevgenia Kozorovitskiy |

The funders had no role in study design, data collection and interpretation, or the decision to submit the work for publication.

## Author contributions

YK, Conception and design, Acquisition of data, Analysis and interpretation of data, Drafting or revising the article; RP, WW, Acquisition of data, Analysis and interpretation of data, Drafting or revising the article; AS, Carried out a subset of mEPSC recording and dendritic spine imaging experiments in older mice, Analysis and interpretation of data, Drafting or revising the article; BLS, Conception and design, Analysis and interpretation of data, Drafting or revising the article

## Ethics

Animal experimentation: This study was performed in strict accordance with the recommendations in the Guide for the Care and Use of Laboratory Animals of the National Institutes of Health. All of the animals were handled according to approved institutional animal care and use committee (IACUC) protocols (03551) of Harvard Medical Area. The protocol was approved by the Harvard Medical Area Standing Committee on Animals. This institution has an approved Animal Welfare Assurance on file with the Office for Laboratory Animal Welfare. The Assurance number on file is A3431-01. All surgery was performed under isoflurane, and every effort was made to minimize suffering.

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
