## [Decision Letter]

Thank you for submitting your work entitled "Neuromodulation of excitatory synaptogenesis in striatal development" for peer review at *eLife*. Your submission has been favorably evaluated by Eve Marder (Senior editor), a Reviewing editor (Marlene Bartos) and three reviewers, one of whom has agreed to reveal their identity: Richard Palmiter.

The reviewers have discussed the reviews with one another and the Reviewing editor has drafted this decision to help you prepare a revised submission.

Summary:

Your study was intensely discussed by the Reviewing editor and three reviewers who came to the conclusion that your manuscript has the potential to be published in *eLife* after major revision. Reviewer 1 stated that the findings only show that neuromodulation can transiently enhance synaptogenesis but that the experiments do not demonstrate that the dopamine effect is 'necessary' for synaptogenesis. Thus, conclusions such as "general role for neuromodulators acting on Gα_s_-coupled GPCRs during neural circuit development would transform our understanding of core principle in developmental neurobiology" (in the Discussion) should be toned down throughout the manuscript. Moreover, reviewer 1 was asking whether a critical time window exists during which the observed phenomena exist. Reviewers 2 and 3 expressed interest in the main findings of the manuscript, in particular that D1Rs and A2aRs can specifically enhance synaptogenesis in dSPN and iSPN, respectively. However, they ask for additional in vitro experiments which demonstrate that the effects of D1 or A2aR agonists are specific to the striatum and not caused by alterations in other cortical areas upon in vivo systemic administration of drugs which reduces/block synaptogenesis. They formulated some key experiments which are attached below. The reviewers and the Reviewing editor felt that these control experiments are critical to support the finding that the observed effects reflect specific actions on iSPNs/dSPNs, but that these should not take more than a month to conduct once the animals are ready.

*Reviewer #1:*

1)The authors demonstrate transient effects of D1R activation on synapse function and locomotion (Figure 1—figure supplement 2), but they never demonstrate that these transient effects translate into any long-term changes in synapses or behavior. The details for Figure 1—figure supplement 2 are not included, but I assume that pups (about P15) were treated with SKF and locomotor activity was assayed for the 30 min following drug administration. It would be interesting to know if a subsequent injection of SKF a day or two later would reveal a more-lasting effect on locomotion, suggestive of a stable change in synaptic strength.

2) As the results stand, the observations described do not establish that dopamine is critical for synaptogenesis on SPNs, but rather they only show that a D1R agonist can transiently bolster synaptic strength in mouse pups that already have normal dopamine signaling.

Reviewer #2:

The authors present a number of interesting experiments that together provide strong evidence for a role of both corticostriatal activity and PKA signaling in SPNs for synaptogenesis. The evidence for spinogenesis in slices is very strong, and the optogenetic and chemogenetic experiments provide supporting evidence for this scheme in vivo. The optogenetic and chemogenetic experiments provide additional evidence for a corticostriatal interaction. Furthermore, the in vivo pharmacology is entirely consistent with the proposed mechanism. In general, I have relatively minor concerns.

1) In Figure 1—figure supplement 1, the authors show the effect of SKF81297 on mEPSC frequency, but don't show the amplitude data. This should be added.

2) Related to the last point, is there a reason why the authors rely on mEPSCs in Figure 1 vs evoked EPSCs in Figure 3 and Figure 4? Evoked EPSCs are relevant, but do not as readily allow one to argue for synaptogenesis vs increased synaptic strength.

3) In light of the experiments from Figure 4, is there a reason why the authors didn't try to optogenetically stimulate cortical inputs in the presence of SKF81297 in slice for 1 hr to elicit an increase an innervation? Demonstrating this mechanism in a reduced preparation (as in Figure 2) provides confidence that the phenomenon can be reduced to cortical afferent activity and PKA activation. I don't see this as essential, but it would strengthen their hypothesis.

*Reviewer #3:*

The paper by Kozorovitskiy et al. follows earlier work by these authors showing that inactivation of dMSNs and iMSNs leads to opposing changes on synapse number in the striatum. That paper, along with others from the Sabatini lab have argued that synaptogenesis in the striatum depends on recurrent feedback from the cortex. Here, the authors argue that in addition to excitatory cortical feedback, synaptogenesis depends on neuromodulation, specifically activation of Gα_s_ by D1Rs or A2aRs in dMSNs or iMSNs, respectively. The idea that neuromodulation gates the ability of cortical feedback to drive synaptogenesis is interesting, as is the possible dependence on Gα_s_.

1) All of the experiments have been done by systemically administering drugs to block synaptogenesis in vivo. Thus, it is unclear whether the effects specifically reflect actions within the basal ganglia. When drugs act to block synaptogenesis, they could be doing so by reducing cortical activity through actions in the cortex or thalamus.

a) To resolve this issue, the authors should ideally perform experiments in striatal brain slices with the cortex cut off. They could use ChR2 stimulation of cortical terminals to elicit excitatory cortical feedback in the presence/absence of drugs to activate/block D1Rs or A2aRs, then measure the effects on synaptogenesis physiologically (by measuring changes in light-evoked EPSCs) or morphologically (by examining spine density). Another possibility would be local infusion of these drugs into the striatum but this is slightly less preferred because there could still be indirect effects of such a manipulation on cortical activity – it would be ideal to have separate control over cortical activity and striatal neuromodulatory state as would be possible in a slice preparation.

b) In order to truly test the hypothesis that D1R or A2aR activation are specifically necessary in dMSNs vs. iMSNs, the authors should also examine the effects of antagonizing D1Rs or A2aRs separately, and also separate out recordings from dMSNs and iMSNs, rather than lumping all four conditions (2 cell types x 2 receptor antagonists) together as has been done in Figure 4.

---

## [Author Response]

Reviewer #1:1)The authors demonstrate transient effects of D1R activation on synapse function and locomotion (Figure 1—figure supplement 2), but they never demonstrate that these transient effects translate into any long-term changes in synapses or behavior. The details for Figure 1—figure supplement 2 are not included, but I assume that pups (about P15) were treated with SKF and locomotor activity was assayed for the 30 min following drug administration. It would be interesting to know if a subsequent injection of SKF a day or two later would reveal a more-lasting effect on locomotion, suggestive of a stable change in synaptic strength.

Thank you for this suggestion. We have completed the proposed experiment, at the ages that matched previous experiments in our study. As was anticipated by the reviewer, we observed that priming with Drd1 receptor agonist SKF 81297 translated into a heightened response to a second dose of the agonist 48 hr later. These observations point toward a lasting effect of a single stimulus enhancing Drd1 receptor signaling. Figures, Methods and Results sections were updated.

*2) As the results stand, the observations described do not establish that dopamine is critical for synaptogenesis on SPNs, but rather they only show that a D1R agonist can transiently bolster synaptic strength in mouse pups that already have normal dopamine signaling.*

We believe that the experiment in Figure 4 addresses this point. We found that in the presence of blockers of Gα_s_–coupled receptors expressed by spiny projection neurons, optogenetic stimulation of glutamatergic projections was no longer sufficient to strengthen light-evoked evoked responses in vivo (Figure 4). This result indicates that Gα_s_–coupled receptor signaling (although not necessarily dopamine specifically) is necessary to strengthen excitatory synapses onto striatal spiny projection neurons.

Reviewer #2:1) In Figure 1—figure supplement 1, the authors show the effect of SKF81297 on mEPSC frequency, but don't show the amplitude data. This should be added.

Thank you; we have added the requested data, updating the figure and text.

2) Related to the last point, is there a reason why the authors rely on mEPSCs in Figure 1 vs evoked EPSCs in Figure 3 and Figure 4? Evoked EPSCs are relevant, but do not as readily allow one to argue for synaptogenesis vs increased synaptic strength.

We believe that the study is strengthened by including multiple metrics related to synaptogenesis and specifically designed the experiments to consider miniature responses, dendritic spine density, glutamate-evoked spinogenesis and optogenetically evoked cortico-striatal responses. Including evoked responses is critical, since an increase in the number of functional excitatory synapses should translate into a greater corticostriatal response – a more direct measure of an increase in connectivity, compared to miniEPSCs and dendritic spine density combinations.

3) In light of the experiments from Figure 4, is there a reason why the authors didn't try to optogenetically stimulate cortical inputs in the presence of SKF81297 in slice for 1 hr to elicit an increase an innervation? Demonstrating this mechanism in a reduced preparation (as in Figure 2) provides confidence that the phenomenon can be reduced to cortical afferent activity and PKA activation. I don't see this as essential, but it would strengthen their hypothesis.

We thank the reviewer for suggesting this experiment, which directly tests the cooperative action of PKA signaling and glutamate release in strengthening glutamatergic connections onto spiny projection neurons. Our results, described in a new figure and reflected in text, demonstrate that indeed, cortical afferent activity and PKA activation are sufficient for inducing this form of plasticity.

Reviewer #3:in vivo

We completely agree with the reviewer that drugs acting to alter synaptogenesis should have circuit level consequences. In order to demonstrate that intra-striatal interactions between cortico-striatal glutamate release and PKA activity are sufficient to rapidly enhance connectivity, we have now performed plasticity induction in the acute slice, though the combination of optogenetic glutamate release and PKA activity stimulation. Figures and text were updated accordingly.

*a) To resolve this issue, the authors should ideally perform experiments in striatal brain slices with the cortex cut off. They could use ChR2 stimulation of cortical terminals to elicit excitatory cortical feedback in the presence/absence of drugs to activate/block D1Rs or A2aRs, then measure the effects on synaptogenesis physiologically (by measuring changes in light-evoked EPSCs) or morphologically (by examining spine density). Another possibility would be local infusion of these drugs into the striatum but this is slightly less preferred because there could still be indirect effects of such a manipulation on cortical activity – it would be ideal to have separate control over cortical activity and striatal neuromodulatory state as would be possible in a slice preparation.*

In order to test whether changes in excitatory synapses on spiny projection neurons are driven by mechanisms within the basal ganglia, we attempted to directly induce synaptic strengthening in the acute coronal slice, which does not preserve most glutamatergic projections into the striatum. Compared to optogenetic stimulation of glutamatergic axons alone, the combination of PKA activating drugs with optogenetic stimulation, induced plasticity, as is evidenced by changed in light-evoked EPSCs. Stimulation of PKA activity alone was insufficient. These data, described in a new figure and text, remove potential confounds of recurrent cortical feedback.

b) In order to truly test the hypothesis that D1R or A2aR activation are specifically necessary in dMSNs vs. iMSNs, the authors should also examine the effects of antagonizing D1Rs or A2aRs separately, and also separate out recordings from dMSNs and iMSNs, rather than lumping all four conditions (2 cell types x 2 receptor antagonists) together as has been done in Figure 4.

We agree that this is an interesting experiment and we hope to do it one day. However, realistically it would take at least 1 year to do so. The experiment requires that we analyze mice with 3 alleles (Cre, floxed ChR2, cell marker) with comparisons across hemispheres (stimulated and control) and two cell types (marker and non-marker). Because of the strong dependence of corticostriatal transmission on litter size (documented in our previous studies), it is necessary to perform all comparisons in each mouse. Thus, we need to try to get sufficient data in each mouse to compare 4 conditions (2 hemispheres x 2 cell types). Then we need to repeat this in control, D1R antagonist, A2A receptor antagonist, and possibly dual D1R/A2A antagonists treated animals (the latter to have a complete block). In total this experiment requires measurements from 4 cellular conditions in 4 pharmacological conditions in 2 different genotypes, each of which includes 3 transgenes. Each experiment also includes acute surgical implantation of a head mounted LED in a young pup and optogenetic stimulation – these are not easy manipulations and are relatively low yield. We hope that the referee agrees that this immense amount of work is beyond the scope of this study and, when considered in conjunction with the rest of the study, is not necessary to support its main conclusions.